# Network Temporality in Percival Everett's Poetry

**Zach Linge**

English Department, Meredith College, Raleigh, NC 27607, USA; zlinge@meredith.edu

**Abstract:** Drawing on new media scholarship, the article suggests that Percival Everett's poetry can be understood through the lens of hypergraphical knowledge. In this context, Everett's poetry operates as a synchronic and diachronic exploration of poetic movements, genres, forms, and inheritances, embodying network-temporal relations similar to the hypernarrator(s) of his fiction. Ultimately, this analysis observes the expansive and cohesive nature of Everett's work, inviting readers to refocus their attention on the indeterminate surface of, and the intricate web of meaning in his poetry.

**Keywords:** poetic form; temporality; literary tradition; postpostmodernism; Percival Everett; new media; hypernarrative; temporal structure; hypergraphical; hypernarrator

## 1. Introduction

To write about lyric poetry is to enter from the outset an impossibly contested territory. Competing historiographies variously emphasize temporal form, for example, or they focus on agents of power who define and contribute to the lexicon. The word "lyric" is itself a field of warring definitions: Scholars and poets argue that the lyric may have simultaneously everything and nothing to do with a writer, chorus, or bard, everything and nothing to do with temporal structure, and everything and nothing to do with genre- or movement-based traditions or inheritance. It would seem, then, difficult at best to approach an author whose work is itself always already engaged with a wide breadth of critical theories and foci and to attempt to make sense of, categorize, or otherwise qualify that work. When the author in question has published somewhere around thirty books of genre-defying and -eluding works of creative writing—five or six of which engage with or are poetry (Can this sentence even continue if we don't know how many of the author in question's works are actually poetry [as opposed to works that *are* not but do *engage with* poetry]?)—the proposition that a single critical lens might encompass or help make some singular sense of that author's work seems at its outset, at best, reductive and, at worst, failed from the start.

The author that this essay considers is, as the title indicates, Percival Everett, whose *oeuvre* eludes easy categorization—which might be to severely understate an argument: as Anthony Stewart writes, Everett's works are "more an attempt to hew out and work in spaces beyond conventional literary categories, not to provide answers but to provoke better questions" (Stewart 2013, p. 216). Everett's genre-elusiveness, when considered in conjunction with how prolific he is, makes sense of the difficulty of stating simple facts about his work. As one example of this difficulty, some interviewers credit Everett with having written six collections of poetry when in fact, he has written only five to date. The sixth book, *There Are No Names for Red*, which does not consist of Everett's poetry, contains Everett's paintings. To complicate matters further, another of the six books, *The Book of Training by Colonel Hap Thompson*, is more of a postmodern instructional in the lineage of Virgil's *Georgics* or Henry Reed's *Lessons of the War*: poetry, yes, but unique even within Everett's own body of already genre-disrupting lineated writing—so unique among Everett's five collections of poetry that it falls outside the scope of the present essay. The four collections that come closest to easy categorization as lyric poetry are themselves so rich with slippage, irony, play, and other hallmarks of (post-[post-])modernism that—to

read them independently from his broader corpus of *non*-lineated creative works—a reader may feel as if they need to be initiated or otherwise instructed how to read the poems. And this is already the challenge with much poetry of the twentieth and twenty-first centuries, that the "difficult poem has created distress for both poets and readers for many years" (Bernstein 2011, p. 3). So, when poetry (which is itself already difficult to read) comes from Everett (whose writing eludes easy categorization), the results can be intimidating.

New media scholarship may, however, provide a useful metaphor and critical apparatus for considering both the difficult expansiveness of Everett's work and its cohesiveness. After all, if we take the author at his word, Everett has something of a unifying goal for his fiction, which is "to suggest that we refocus our gaze from the transcendental connections of meaning(s) toward the obscure and indeterminate surface of fiction" (Everett 2004, p. 154). This essay observes how the same suggestion applies to temporal structure in Everett's poetry, which both works within the constraints of language, thereby structurally signaling "connections of meaning(s)", and draws attention to craft, language, and genre as both "obscure and indeterminate". In her 2013 essay, "Everett's Hypernarrator", Judith Roof argues that the narrators in Everett's novels emerge as a "hypernarrator" that is "produced as an effect of a narrative dynamic that has abandoned the linear in favour of a practice of association, possibility, multiplicity, and polysemy" (Roof 2013, p. 214). Everett's poetry embodies and enacts a similar hyperreality. Rather than providing for the emergence of a unifying "hypernarrator", however, Everett's poetry is closest to what Alan Liu calls "hypergraphical" knowledge, which "is oriented toward multimode, multivectoral, and multiconnection graphics representing . . . aggregate perspectives" (Liu 2018, p. 71). The poetry is at once synchronic in its affiliation with a named author—however complicated the figure of the author might be, of course—and is diachronic in its vast and varied engagement with poetic movements, genres, forms, and inheritances from written poetry's nascence to the present day. In short, Everett's poetry functions along hypergraphical, network-temporal relations that are inherently and conspicuously referential at multiple levels of scale.

## 2. Sacred and Profane Time in Network

In *Friending the Past: The Sense of History in the Digital Age*, Liu argues that the "sense of history across ages and media" is akin to an archeologist's "stadial" categorizations (Liu 2018, p. 60), as the act of dividing present and past knowledge at first appears linear. Liu comes to see this same sense of history "as an example of the construction of a 'new media encounter' differentiating old and new stages of media on a modernizing line of change" (Liu 2018, p. 60). The new media encounter, Liu says, has been and continues to be a failed attempt to move beyond linearity. Similarly, in his study of temporality in the poetry of Felipe Fortuna, Antonio Rediver Guizzo observes a specifically Judeo-Christian branch of historical-religious perspectives that are guided by a "mitologia de orientação progressive do tempo" (Guizzo 2016, p. 969). This transnational branch of thought consists of mythologies driven by paradisaical ideals. Its adherents both wish to return to an imaginary and pure, earlier state and also to be released from the constraints of linearity. Linearity, within these mythologies, may be associated with progress, but the linear movement is oriented specifically against and away from an ideal state of innocence situated toward the beginning of time, with the seemingly unrealizable alternative goal being to escape time altogether. Linear mythologies, then, are always already failed in their incapacity to capture a reality that operates both linearly and nonlinearly.

Guizzo, however, also observes two transnational literary ideals that contrast Judeo-Christian mythologies of temporal constraint. On the one hand, there are the models by which so-called "*religiosas orientais*" narrate time according to the circularity of eras; this form of narration rejects the notion that an individual is a historical being and emphasizes era, archetype, and emergence over progression (Guizzo 2016, p. 969). On the other hand, Guizzo observes a transtemporal figuration of poetry that transcends the aesthetics and culture of the moment in which it originates (Guizzo 2016, p. 966). Poetry transcends,



Guizzo says, as it arises through its attachment to passion and its "*traduzir/transfigurar*" of the relationship between humankind and the world (Guizzo 2016, p. 966)—if we take "the world" to mean, generally, reality. When we reconsider, in this context, the section of Liu's text so far discussed, we see support for his proposal that "we cannot be emancipated into postlinearity" (Guizzo 2016, p. 61), only insofar as our figure for conceiving time follows one vector. However, we also see millennia of precedence for imagining and constructing language that mimics multivectoral nonlinearity.

Everett writes in the first and eponymous poem in *Trout's Lie*:

> the full sun complete surface,
> i sol tace,
> streams along, lap-lapping
> against itself, against its current,
> against its own origin,
> where dim hours wait (Everett 2015, p. 11)

For Everett, even the "complete surface"—which is figured here as the "full sun"—has an origin, and in its origin, time is both dim and anticipatory. The complete surface of the sun, however, reflects itself in a river, whose lapping against the bank is, too, a reflection of the sun and its gravity. The surface is suspended in space. It seems complete in itself, yet it also proves incomplete in that the surface reflects itself in the metaphorical river, makes waves from its own pull, and as such, moves both outward and back toward itself, through itself. If, for Everett, the surface is both complete and incomplete, so, too, for Liu, does the vector seem insufficient when taken alone. Rather than turn to circular figurations to remedy this metaphorical insufficiency, Liu admits that even his "historical narrative about linearity [is] a fairy tale . . . because it doesn't really bear up to close inspection" (Liu 2018, p. 79); he then mines new media for alternatives to his own narrative.

Liu finds an alternative in digital literary modes, such as Franco Moretti's distant reading, which "sample, modularize, encode, and otherwise transform different media into a fungible common stock of data and metadata, thus facilitating the visualizing of text" (Liu 2018, p. 62). With these visualizations, what "stands 'revealed' . . . is today's dominant mode of graphical knowledge, which I loosely call *hypergraphical* (formed by analogy with 'hypertextual')" (Liu 2018, p. 70). The hypergraphical synthesizes aggregate data into visual knowledge formations that emphasize relations between the individual datum and other data. It operates closer to what Guizzo calls "sacred" time (cyclical and reupdatable), as opposed to "linear" time (progressive), in that the visual (hyper-)graphic emphasizes complex associations and relations between data. Importantly, the graphic's viewer apprehends these data differently depending on the scale at which that viewer operates. The simple exercise of "zooming in" on any visualized and graphed fractal equation, for example, would show complete images no matter the scale, thus emphasizing the complexity of a given subset of the data visualized and also drawing attention to the complexity of the graph as a whole.

Liu says hypergraphical knowledge operates in networks. Such knowledge "seeks to keep in view both atomistic node-entities in their local contexts and the overall relationality of the whole graphical construct" (Liu 2018, p. 71). Hypergraphs' visualizations extend beyond the single vector of linear thinking in a mode more similar to the circular and emergent; they are "not just *hyper-*, but also *infra-* and *super-*, or *micro-* and *macro-*, graphical" (Liu 2018, p. 71). They not only visualize the node—or, in this discussion of Everett, poetic utterance—in its present moment, but they emphasize its relations to other nodes, thereby evidencing each node as connected to its graph. Though ultimately apprehended linearly, the nodes' and poems' material qualities, compositions, groupings (or genres), and temporal structures point beyond the individual node, beyond the individual poem, and toward the graphic network, be it strictly hyper in the internet sense or resituated in physical literature.[1] Nodes, in their being identified as such, suggest in their constitution a consideration of scale and value. They operate temporally both as singular nodes apprehended profanely—

within the confines of linear reading—and through their drawing attention to the broader, sacred, referential network.

*Ars Poetica*

Consider the surface of the sun: its immense heat, its solar flares. It envelops enough energy and mass to drag planets through space, and planets' seasons arrive and pass according to their turns in relation to it. On Earth, tides exist due to the draw upon the water's surface from the sun and its smaller celestial sibling, the moon—both bodies that humankind has thought of as gods; we've named kings after the sun and moon, and days of the week. In the Everett poem previously discussed, the sun's royal name is "full sun complete surface". The celestial body is not merely its name but its "full" name; the body is not one characteristic of the sun, but the totality of its being—"full" sun. It is not only its surface, but it is also its physical relationship to water on Earth, as seen in the waves "lap-lapping" against themselves in a moment of juxtaposition and conflation between the sun's flames and water's waves. As the sun's surface surfaces in this poem, it is yet again figured metaphorically, but instead of a god, its figuration is the "complete surface" of a total, all-encompassing thing phenomenologically set in relation to itself. What is the sun? Surface. What is water? Surface in waves. Where does each exist? At a distance from each other and as one and the same.

Surface emerges as one of several considerations in Everett's poetry, which takes familiar object images, like the sun, and reanimates them in new figurations. The surface of the sun, for example, appears suddenly mathematical—like a graph or an equation for the physics of a wave—and it simultaneously highlights the surface of language for its inherent slipperiness and for our slippages in reading, speaking, and thinking. Slippage itself is yet another consideration of Everett's poetry: when meaning breaks down through repetition, as in "The dead are dead as dead and deader" (Everett 2015, p. 19); when the tools we use to communicate are not positioned in their accustomed situations, such as "dreams in a paper sack" (Everett 2015, p. 18); or when those tools or objects are made to stand in for something, as in the metonym: "The tavern's just a table leg" (Everett 2015, p. 48). So much surface on the sun provides the poet with so many ways to play with representation. As per Erich Auerbach, figurations (or "figura") are "derived from the stem" of the word "and are not" (Auerbach and Valesio 1984, p. 11); they hearken back to the linguistic molds that made them and can also veer into "purely abstract meaning" (Auerbach and Valesio 1984, p. 14). They can, in short, both connote a plastic, ur-form conceptually similar to the linguistic signified and can be intended to mean nothing more than the concrete visual shape and shared sounds of the spoken, abstract signifier. The word opens outward in Everett, as elsewhere, as it delimits possibility, and in this way functions on the level of scale.

But however many directions one word can point, there is a limit to what one writer can signify. Because of how prolific he is, Everett's genius may seem to the reader never-ending, but as with all artists, the scope of his work is apprehended and approachable by a limited number of preoccupations. Not so limited are they that I or—I imagine—any writer would attempt to catalog each of them, but these preoccupations are at least related closely enough that even Everett writes explicitly about them as if to say: "This is *how* I am discussing *what* I am discussing. Read". This section looks at some examples of these foregrounded topics by reading them within the genre of *ars poetica*. We'll take *ars poetica* to mean a "poem that explains the 'art of poetry', or a meditation on poetry using the form and techniques of a poem" and will ignore, as secondary to our consideration, the "modernist *ars poetica* poets [who] argue that poems should be written for their own sake, as art for the sake of art" (Greene 2012, p. 1). Such explanations and meditations pervade much of Everett's poetry, from his first published collection of poems, *re: f (gesture)*, through his fourth, *Trout's Lie*.

The first example of an Everett *ars poetica* is also an abecedarian, which is a poem that conventionally consists of the same number of stanzas as there are letters in whatever

language the poem is written, and in which each stanza begins with the next successive letter of the alphabet until the end of the alphabet is reached. The earliest known examples of abecedarians are Semitic, from Hebrew religious poetry (Greene 2012, p. 1), and the form is contemporarily used both by poets and for word games in classrooms. The form draws on a long history as both sacred and playful, an act of worship and a game. Everett's abecedarian—"Zulus" from *re: f (gesture)*—immediately draws on tradition:

> A is for Achitophel.
> It was he who put Absalom
> up to the big naughty.
> Dryden called Achitophel
> a great wit. Not to
> Blow Dryden off, but the
> wit was Solomons's. (Everett 2006, p. 15)

Everett begins his first book of poems with an allusion to John Dryden's satirical book-length poem, *Absalom and Achitophel*, and in doing so, he creates an immediately intertextual space on the page. In this space, Everett's poetry is in conversation with poetry that is itself already in conversation with other texts.[2] Dryden's text, published in 1681, writes over and into a story that was written roughly two thousand years previously (Britannica 2013). Dryden's satire picks up the millennia-old story to write a versified, satirical commentary on the politics of his time, and Everett picks up the same story 300 years after Dryden to also comment on the politics and poetry of his own time, but not before engaging the source(s') authors. This tradition of writing into and on top of inherited narratives evokes a sort of palimpsest, where the materiality of the poem as an object that changes with time is matched by form in its content: "Not to/Blow Dryden off", Everett writes, indicating both the use of source material and that material's reformation.

In this first section of the poem, Everett calls Solomon "small/and a little queer" (Everett 2006, p. 15); the latter term I take to mean generally "odd" rather than specifically non-heteronormative, although perhaps there is no need to rule that possibility out. By referencing Solomon, Everett begins to construct a transhistorical lineage through which to trace the poem's topical considerations. He also name drops Aristotle ("A is for Aristotle/who learned from Plato") and Anaximander, "who/said that the element of/things is Boundless" (Everett 2006, p. 15). Everett capitalizes "Boundless" as a Romantic poet might capitalize "Beauty" or "Truth," and he uses both the lower and upper cases for the first letters of the section's lines. The inconsistent use of capitalization might embody a nod to both traditions in which every first letter of the line is capitalized and more modern and contemporary poems in which the line (and its first letter's capitalization) loses its primacy. The poem is lineated, yes, but it also points to the line and its capitalization as two interchangeable and changing formal elements of poetry. So, too, does Everett consider genre for its permeable borders, in this poem that is at once an abecedarian, a list of literary and historical figures, and an *ars poetica*.

Having listed some sources and commented on each, Everett ends the first section of "Zulus" with his own commentary on attributed statements: "the element of/things is Boundless" (Everett 2006, p. 15). The poem tells its readers, through sourced material, that its material, form, and content open upward and outward. This will be, the poem appears to say, not the last word on the subject but another word opening outward to the past and to the imagined future as it opens upward to a possible reader's interventions. Or, as Michael John states: "Lyric time and poetic engagement encounter each other in the future reader's potentiality" (Michael John 2017, p. 271). How, then, is this poem an *ars poetica*? It uses the form and techniques of a range of poetry to meditate on what a poem is: its inheritances, its speaker, its composition on the page, its readers, and how those readers (might) engage. It is precisely this *writing about poetry*—as opposed to writing about, for example, love or war—that makes this poem an *ars poetica*.

Let us consider the poem's relationship with the temporal structures of other abecedarians. Much of "Zulus" appears to operate from an atemporal perspective in that it makes

rhetorical assertions, and in the way, those assertions are linked outside of time to other events: the past tense, for example, in "Dryden called Achitophel", and the transtemporal rendering of wit in the passive-voiced clause, "Not to/Blow Dryden off, but the/wit was Solomons's" (Everett 2006, p. 15). Tania Notarius observes that biblical poetry "employs available morpho-syntactic and pragmatic indications that facilitate an adequate temporal interpretation of most sentences" (Notarius 2011, p. 280). Everett's syntactical indications, quoted earlier in this paragraph, operate the same way that cross-clausal temporal anaphora operate in Biblical Hebrew. Those of Everett's poems that "are primarily lyric meditations" are not, as Helen Vendler says they often are, "phrased in the present tense alone" (Vendler 2010, p. 112). Rather, they "represent bounded events", in that Dryden *called* and the wit *was*—which is to say that both events (calling and being) happened in writing, and therefore temporally unbound, but because both writing events "represent bounded events", i.e., historical events, "they are located in the past and get an episodic interpretation" (Notarius 2011, p. 285). Because the rhetorical assertions contained in "Zulus" are writing events, their present-tense assertions about Dryden and Solomon "are interpreted in the past due to the temporal inference of local phrases; they create the necessary temporal anaphora for the temporal interpretation of verbal statements in the matrix clauses" (Notarius 2011, p. 285). As the Biblical Hebrew of, among other examples, the Psalmist's abecedarians might refer to "bounded events" within anaphoric time, so, too, does "Zulus" reference a litany of events without restricting them to a strict past. This is the temporal plane of allusion within which "Zulus" makes its arguments about what poetry is and can do: one that appears because of its linguistic structure as situated both within and outside of linear time.

If, however, we are to keep with Guizzo's characterization of Judeo-Christian poetics and narratives as belonging to a mythology of progressive time, we need to consider content as well as additional temporal forms in Biblical Hebrew. It would seem, based on our discussion of anaphoric atemporality, that Hebrew religious poetry operates outside of time, but it is deictic and sequential time that characterizes much Biblical poetry. Sequential time, for example, "builds an autonomous temporal succession of events/situations, usually in chronological order" (Notarius 2011, p. 277). Deictic time "establishes reference to" speech time and, as such, is concerned with past speech acts even when written in the present tense. What makes anaphoric time unique is that it "refers to another, non-[Speech Time], contextually established" reference time (Notarius 2011, p. 277) and that reference time is constructed by a network of transtemporal relations, as opposed to a strict affiliation with a speech act or event. The abecedarian thus emphasizes argument over event through its transtemporality. It does not rely on a chronological presentation of events, and it does not limit itself to speech time.

Where content is concerned, the argument in "Zulus" reaches quickly beyond so-called "Western" influences on art and poetry. If the first section traces episodic retellings of a story from a Biblical inheritance through a Western poet to Everett, the poem's second section moves geographically elsewhere. Specifically, section two moves to an event in South Africa, "where/three Boers were slightly/wounded on 16 December 1838" (Everett 2006, p. 16). It continues: "Three hours of battle,/leaving three thousand Zulus dead" (Everett 2006, p. 16). The second sentence's shift from the past tense to the continuous present tense argues that there is a structural difference between the Boers and Zulus: three Boers *were* wounded and on a specific date, whereas that same "battle" is portrayed as *leaving* three thousand Zulus dead. The relationship between this poem's form and its content is evident in this second section, in how both work in tandem. The present continuous tense speaks to an ongoing battle for some, and the past tense minimizes others' suffering by restricting it to a moment in the past. The three thousand Zulus' deaths continue; the three Boers' wounds have passed. The abecedarian's form draws structurally on multiple, sometimes apparently contradictory, registers of time and, in so doing, provides a literary space where the historical "past" can be both past- and present-tense. Simultaneously and similarly, the poem's content highlights lived realities that appear to contradict one another:

both the Boers and Zulus, historically, "were" wounded, but only one of these populations continues to experience a past-tense event in the present moment.

Everett's use of the anaphoric atemporal perspective in "Zulus" situates global, transtemporal speech acts and events in conversation with each other. It constructs an *ars poetica* that enacts his statement of poetics. The poem is palimpsestic in how it writes into, onto, and through events, historical and literary figures, and speech acts. It is also, in this way, operating according to a cyclical, reupdatable logic: event is episodic, thought is nonlinear and transtemporal, and what emerges from the organized chaos of literary influence is a set of formal and topical concerns. If poetry is as chaotic as the universe it inhabits, then language is the mechanism and frame for observing chaos's emergent properties—just as visualized graphs function on levels of scale and intervention. In "Zulus", a primary concern is the data cluster constructed as race. The Boers, colonists of South Africa, "were slightly wounded". In contrast to the past-tense Boers, three thousand Zulus exist in a state of perpetual death because they are not only an ethnic group in South Africa but because they can also signify the racially oppressed elsewhere and at other times. In the third section of the abecedarian *ars poetica* (grouped by the letter "C"), Everett writes, "C is for Chandler, Happy/because he is caucasian", and he leaves "caucasian" in all lower case (Everett 2006, p. 17). Chandler, according to the poem, "sings about 'darkies in the field'/before twenty-three thousand/white faces while black men/wait to play ball" (Everett 2006, p. 17). In the fourth section, Everett alludes to Langston Hughes' characterization of Black Americans' dreams as cyclical: "Dreams are often deferred/spiraling round and round,/creeping through generations" (Everett 2006, p. 18). The remainder of "Zulus" is loaded with literary, mythological, and historical allusions, from Ralph Ellison (Everett 2006, p. 19) to Mary Shelley (Everett 2006, p. 20), from Bonaparte (Everett 2006, p. 16) to Robespierre (Everett 2006, p. 32), from mathematicians G. H. Hardy and Wilhelm Weinberg to philosophers like Immanuel Kant and religious texts like the Qu'ran (Everett 2006, p. 22). Figures and texts surface in what reads as a transtemporal, transnational meditation on mythology, power, language, and race.

Repeated phrases and quoted material are stitched together to build meaning and texture as they respond to, mirror, and depart from other phrases and material. The phrase "Always name offspring", for example, first appears in the sixth section ("F"): "F is for Frankenstein/who did not name his baby./Always name offspring". It then appears in the fourteenth section, after Everett puts two texts in conversation: the motto "*novus ordo seclorum*", from the reverse side of the United States Great Seal, and the phrase "the number of his name" (Everett 2006, p. 28), which is contained in the Book of Revelation as a modifier to "the name of the beast" (Cambridge UP 2004, 13:17). In whole, the section reads:

> N is for novus ordo seclorum,
> that prophetic adornment,
> that frightening revelation . . .
> "the number of his name."
> Always name offspring.
> N is for natural, sharper flats.
> "In music the passions enjoy themselves."
> Nights without melodies
> kill without conscience. (Everett 2006, p. 28)

Charles Thompson joins the Book of Revelation and Friedrich Nietzsche, and meaning accumulates through associations between what each of the alluded-to texts signifies. The poem's argument further accumulates in how these significations are informed by the poem's overarching concern with power and race. In Everett's first *ars poetica*, the alphabet is the organizing principle and meat grinder. The poem's speaker collects quanta and stitches it together to create new meaning. Whatever concern emerges in the poem does so by way of that collected quanta and how it creates meaning through its being stitched into the poem.

Writing at length about "Zulus" might show how many texts and ideas are at play in Everett's poetry, as is the case with much of Everett's writing, but it should also indicate that—despite the poem's breadth of referentiality—there is at least one primary argument in each of Everett's poems. In "Zulus", one argument, when rendered prosaic, might reductively read: Power is episodic: one (ethnic) group exploits another (ethnic) group. But Everett is not concerned exclusively with argument, so much as he is concerned with both arguments and how arguments are made. He foregrounds the alphabet as an organizing principle in this first poem, and he nods to the alphabet as a constructive power in the second poem of the collection, "The Hyoid Bone": "Brace the words, the delicate instrument,/the tongue for sweet kissing, upsilon" ([Everett 2006](), p. 43). As *re: f (gesture)* departs from *ars poetica* and moves into an ode, the first stanza of the second poem ends with the twentieth letter of the Greek alphabet, upsilon. What's more, this letter (and the ode) foregrounds Everett's interest in language as a cite of pleasure *and* pain. Upsilon—rendered in the English upper case as the letter "Y"—is equated to the tongue. If it is a tongue, however, based on its shape, it is forked. "The Hyoid Bone" concludes: "Fracture this bone, compromise the support,/and feel the true anguish of speech" ([Everett 2006](), p. 43). Although Everett has moved from *ars poetica* to ode, the book reveals that a central concern of the first poem—argument and the linguistic construction of argument—will remain a central concern in poems that follow.

To some readers, it may seem that a difficult poem can be interpreted in a variety of ways. And while it is true that in "Zulus", the quanta that Everett and the poem's speaker stitch together operate on a seemingly limitless plane of associations, the poem's meta-argument is not so flexible. "Zulus" may be "about" power, race, and many other subjects, but it is as much an argument about the linguistic constructedness of those subjects and of poetry itself. If we doubt that this is a primary concern of Everett's based on "Zulus" alone, we have that doubt dispelled by many of Everett's other poems. Take, for example, the first section of "Short Circuit" from Everett's second book of poems, *ABSTRAKTION UND EINFÜHLUNG*:

> Eradicate the boundaries, obscure the edges,
> collage, montage, assemblage, flying
> in the face of the housing structure,
> seeking at once inclusion and acknowledged exit.
>
> The building has no permanence, the concept
> of the building has no permanence, only
> the event of the art, the ephemeral moment,
> only itself, stealing from itself, from himself
>
> and three others, the final illusion being
> that any of it at all is ready-made.
> That fuzzy, blurry, unfocused gaze on a
> world personal yet never personalized. ([Everett 2008](), p. 57)

The second person is implied in the imperative grammatical mood of the poem's first line, which instructs said addressee to "Eradicate the boundaries" and "obscure the edges", thus rendering the poem's object unbound and, if delimited by extant edges, at the very least resistant to its edges. We do not, however, have an exact object in the poem, so much as another metaphor: the "building" stands in for, it seems, anything made. Nor do we have an exact second person, which seems appropriate when we read the last line: the world is "personal", yes, but it is "never personalized". So, though there may be an implied second person in the direct address of the first line, the addressee does not appear to be any one individual, so much as the implied "you" can be any person who takes part in conceiving of, entering, or seeking an exit from the "building." Further, that building is equated to something. One might expect an objective correlative from a Romantic metaphor, in which the imaginary abstract is tied to the concrete. Or one might expect metaphor to assume

"both a structural and a mimetic function for poetic language" in the Aristotelian tradition (Hamlin 1974, p. 172). Everett's metaphor, however, operates in the opposite direction from a Romantic metaphor and does not require a signified upon which to base mimesis: the building is as close to abstraction as the idea to which it is compared, which is "the event of the art". The poem further describes the building or a concept of it, and the event of the art, or an idea of art, as "That fuzzy, blurry, unfocused gaze on a/world". Those who encounter a building or art event are incapable of perceiving either building or art but through an "unfocused gaze". Not only that, but the world they perceive is not even "the" world in a singular sense but "a" world. The poem refuses to reduce any idea or event to the definite article; the indefinite article reminds readers that ideas, events, and even this poem itself are constructed things poorly perceived by people having personal experiences in a never-personalized and finite reality. To personalize would be to conflate the person with what said person perceives.

### 3. Artifice

"Zulus" and the first section of "Short Circuit" show a poetics that is encyclopedically concerned with a range of source material, and Everett's poems' speakers evince the author's preoccupation with language—how it is constructed, how it constructs meaning, and how these two categories break down. In the following poems, as often elsewhere, Everett foregrounds artifice sometimes to bolster content and meaning but as often to disrupt and delay language's meaning-making. Consider the second section of "Short Circuit":

> The building
> has no permanence.
> What else is there
> to say? (Everett 2008, p. 58)

If poetry were concerned exclusively with poeticity, there would be little to say: poems, as both literal and physical constructs, are held by people; they are transient; they change, decay, and ultimately disappear. So, although Everett's poems frequently comment on what poetry is and can do, the *ars poetica* makes for limited and circular poetic fodder. *Ars longa, vita brevis*. If the building has no permanence, however, what the poem points to is not only its subject matter but its relations to other subjects and, as importantly, its relation to being. That there was a building once would have no consequence; that someone made it, however, presents further lines of questions for the poet/poem to consider, such as how, what person(s), with what tools, and toward what aim. The poem's third section answers some of these questions:

> watt
> hells
> is theair
> too
> slay (Everett 2008, p. 59)

The speaker's role in creating the poem is to work with available content and challenge its shape. The poet's role is to recognize the speaker as constructed in and of itself. So, the speaker of this poem repeats itself, but its relationship to language is changed. Be it gimmick or meditation, the adlib's restatement of the second section's final couplet bucks against the finality of its line of questioning. The rhetorical question, "What else is there to say" is answered: "Maybe much of the same, but differently". In this way, the poem takes its same question and reinvigorates that question by providing it with a new shape. Stanley Plumly writes that lyric poems "often resemble shapes in nature, which also create an artifice: the meander, the branch, the elliptical circle, the hexagon within the snowflake or the xylem of a tree, and the spiral or the gyre" (Plumly 2004, p. 269). In "Short Circuit", as elsewhere, the poem's shape changes, though the argument does not.



In toying with shape and disruptions of language's primacy over meaning, Everett opens his poetry to comparison with experimental poetry, from Gertrude Stein to the so-called "language poets" of the late twentieth century. There are additional ways the poems invite such comparison. With "Zulus", we see writing operate as a sort of palimpsest, which Marjorie Perloff says "may well be the poetic form native to" language poetry (Perloff 1998, p. 254). Additionally, language poetry "decidedly shifts the emphasis away to . . . the signifier, the material aspects of the medium, denying the reader the possibility of construing the text as a coherent utterance from a consistent subjective perspective" (Hühn 1998, p. 218). In his first four books of poetry, Everett's speakers are unnamed; the poems operate in or similarly to a variety of inherited forms from the abecedarian to the ode, and they pressure mimesis in each; and a number of poems also veer into being entirely unmoored from meaning. One might read echoes of Stein's "If I Told Him, A Completed Portrait of Picasso" in much of Everett's *Swimming Swimmers Swimming*. Where Stein writes, "Shutters shut and shutters and so shutters shut and" (Stein 2008, p. 190), Everett replies, "Doubting doubts that doubted/doubtless doubting doubts" (Everett 2011, p. 35), and, "never never/never never/never/never/never believe" (Everett 2011, p. 24). In her poem "Sacred Emily", Stein writes, "Rose is a rose is a rose is a rose. [. . .] Pages ages page ages page ages" (Stein 1922, p. 187). In his poem "Rows," Everett writes: "the rose/and the book/are the same color," and he ends the poem: "like the rose is like the book is like the rose is the color of the rose" (Everett 2011, p. 32). In the less precisely locatable intersections of language and meaning that Everett's poetry inhabits, by comparison to Stein, moments of unmoored meaning may leave the reader wanting orientation (as sometimes also happens in his fiction)—and indeed, Stein's poetry similarly challenged expectations for poetry's mimetic and sense-making functions among her readership. Where Everett's poetry is located at an intersection of so many poetic inheritances, however, we see in Stein a particularly cubistic thinking, one that it appears can also inform our reading of Everett, though more limitedly. Cubistic artwork forms its figures through the interplay of various textures, surfaces, shades, shapes, and colors. So, too, do Stein's poems construct their meaning through the surfaces of language and those surfaces' interplay. Everett's consideration of the "book" object alongside the "rose" object in "Rows" seems at the very least a nod to Stein's famous poem, and indeed a previously quoted poem draws attention to techniques that flourished in and, in response to cubism: "collage, montage, assemblage" (Everett 2008, p. 57).

The difficulty in appreciating unmoored poems, then, is not only that they upset expectations for language's meaning-making function but that to appreciate why they upset these expectations, a reader must be initiated into (art and) poetry. This is by no means a new difficulty. Everett alludes to another difficult and allusive poet in his long poem, "Insinuation", from *Swimming Swimmers Swimming*. "Insinuation" begins: "And it starts with a conjunction,/When the night sprawls beneath itself/Like a dream beneath a thought" (Everett 2011, p. 13). T. S. Eliot's "The Love Song of J. Alfred Prufrock" similarly begins: "Let us go then, you and I,/When the evening is spread out against the sky/Like a patient etherized upon a table" (Eliot 1963, p. 3). The parallel between these poems is slant, but similar murmurations continue to suggest that Everett is alluding to Eliot throughout the poem. Everett's "And when they dissect me on that table" (Everett 2011, p. 16) seems to bring readers back to that etherized patient in Eliot's poem, and Everett's "A hundred rewritings and revisits/Before we sit and have coffee" (Eliot 1963, p. 14) might remind readers of the speaker in Eliot's poem having "measured out my life with coffee spoons" after predicting that there will be "time yet" for "a hundred visions and revisions" (Eliot 1963, p. 4). The poem also alludes to Andrew Marvell—"Like some coy mistress/The beige smoke rides" (Everett 2011, p. 14)—and in a poem that alludes to at least two other poets, we might also suspect an allusion to Plath when Everett writes, "Ask any zombie, maybe even Lazarus/himself,/And he will tell you,/Returning from the dead is not/The same as being alive" (Everett 2011, p. 19). Why this difficulty? Why foreground the poem's artifice, its intertextuality? If Everett's poetry wishes to respond to the problem posed in "Short

Circuit"—"What else is there to say?"—it seems his best attempt to locate his response is in difficulty, in the assemblage/collage/montage of sourced material set in conversation with his own wonderings. He continuously speculates about language, ending, for example, the long poem "Insinuation":

> Crippled by indecisiveness,
> Will I ever pass by that wind with no purpose?
> Pragmatic and lost in a sea of doubt,
> Will obscuring meaning be enough
> To find me to the river's edge?
> [...]
> It will be a child's voice
> That leads me brow deep into the flow. (Everett 2011, p. 21)

In a section that requires little explication for its relative candor, what stands out is the question, "Will obscuring meaning be enough". Everett's speaker emerges from an obscure, intertextual assemblage to express doubt, which the poem expresses clearly. But the poem's speaker also seems eager to alleviate doubt, as "Insinuation" quickly concludes with the prophetic couplet: "It will be", stated in no uncertain terms, not the act of "obscuring meaning" but "a child's voice" that solves some metaphorically ambiguous problem. And the poem's problem *is* ambiguous. Readers trained to look for mimesis or objective correlatives have by now begun to expect to have their expectations upended. Similarly, readers may reach the end of this poem and still wonder what is referred to: a problem in the so-called "real" or physical world, a problem with language, or a problem located somewhere else on the unlimited graph across the surfaces of which these poems play?

It is frustrating to read Everett's poems. This aspect of the experience should not be ignored. To suppress that frustration would be to risk losing confidence in the complexity of Everett's overarching *ars poetica*, which says yes, reading these poems is frustrating, but language is frustrating. Language frustrates meaning. Texts are frustrating in that they are too many to track and too varied to siphon into some cloying and singular sense. Further, the content that language tracks and constructs is that of human experiences, which are coded by so many competing types of history—linear, temporal, atemporal, cyclical, and so on—and which all appear, to Everett's speaker, to tell the same stories: of peoples exploiting peoples, and of people in love. This frustration, it turns out, is a constituent part of language itself, as the poem "Grammar" demonstrates:

> and she cries.
> and he cries.
> cries and she.
> he and cries.
> .cries she and (Everett 2011, p. 26)

Everett writes in the poem titled, tongue in cheek, "Cañon", that "There is no house no structure, no question, no answer./There is no house no question no planks no structure" (Everett 2011, p. 33). There is no permanence in these poems. Even the lyric "I" "is all/merely idea anyway" (Everett 2011, p. 49). In pre-modern poetry, as Candace Lang writes, "authors 'of the single' metaphor were so single-minded . . . as to attempt either to suppress their own subjectivity or to deny the existence of the external world," and authors of the "'double metaphor' sought to establish an equilibrium or synthesis between inner self and outer existence" (Lang 1982, p. 2). By contrast, for Everett, the "components" of a self "line/up/like points on/a grid" (Everett 2011, p. 48). The self is a set of data chaotically clustered; however consistent its emergent properties—those points on the grid—may appear.

## 4. Genre Trouble

*Ars poetica* and artifice, allusion and collage—all terms with which literary scholars wish to structurally assort genres are swept up by Everett's poems into an overarching

project, it seems, of simultaneous poetic embodiment and commentary. There is so much more to be said about how these poems frustrate meaning and to what effect—the role of authorship in Everett's fifth book of poems, *The Book of Training by Colonel Hap Thompson* (Everett 2018); how madlibs work throughout *ABSTRAKTION UND EINFÜHLUNG*, for example; how bricolage and patchwork function in *re: f (gesture)*'s "Zulus" or *ABSTRAK-TION*'s "Picasso"; the way slippage punctuates *Swimming Swimmers Swimming*; and how Everett reconceives of the love poem and ode within the limits of his lyric-theoretical framework. What many of these poems have in common, however, is that the boundaries are uncertain and obfuscated between one poem and the next, one genre and another. There may exist, to risk the word, "thematic" concerns that pervade these first four collections. No single concern, however, would exhaust the richly complex, intertextual, and temporally quantum craft of the poems.

If this description taps into a reader's difficulty with describing Everett's poetry, it is a difficulty that is shared by the author. Everett's poems constantly bump up against sudden borders in content and form. "The truth that/we never read, that/we so reluctantly/accept," the poem "Shame" says, "is that there/are no new massacres,/no fresh travesties,/no original genocides" (Everett 2015, p. 37). In a conspicuously rhetorical gesture, the speaker vocalizes what appears to be the poet's own frustration: there is no newness; the story is old, though too infrequently told. The poems continue to prod at the question of solving some ambiguous problem, presumably with form. Can the poem tell the story obscurely enough? Can risking frustration reach the reward of change? And if that change cannot be progress, then can it at least be a change in how the story is told?

In Everett's first four books of poems, these questions are governed by a theoretical logic: "Points point to,/Pinpoint loci/On a dimensional plane" (Everett 2015, p. 52). The plane that these poems describe is itself dimensionless but for the reader's or writer's intervention, but for the scale at which the plane is perceived, and but for how the argument is constructed into time and language. "The argument rings,/Bells on a chime, No more right than/Pound and Eliot", Everett writes: "I know this [...] That there is nothing/More, nothing left of/Auden and Breton" (Everett 2015, p. 15). What it means to write in a relativistic universe, from within the relativistic constraints of time, is that the author must write into and embody contradiction. He must "know" that W. H. Auden and André Breton are both dead and also acknowledge that writing and history do not necessarily operate within progressive, linear (profane) time. They are dead; they live on. Further, the "they" there is itself illusory in that Auden and Breton are figures rendered through language; to consider them metaphorically, their "loci" on the hypergraphical plane has merely been rearranged. Where do they exist? Not here and now, but also here and now; they exist differently.

Hypergraphical knowledge "is oriented toward multimode, multivectoral, and multiconnection graphics representing . . . aggregate perspectives" (Liu 2018, p. 71). There is both the graph and the knowledge mapped out on the graph. The ideas in Everett's poems are both his and others', sourced and untraceable, assembled into writing that explores topics like power, race, love, and language by using multiple genres, temporal structures, and degrees of affiliation with meaning. The ideas are not old but are explored differently, the poems claim, based on how they are written and arranged. Hypergraphical knowledge, however, is defined not only by the knowledge on its planes but by the rules of those planes and the graph on which they are situated. David Baker and Ann Townsend write, "If a poem's linear and bodily forms provide its apparent exterior, then its rhetorical form—the structure of its story, the shape of its argument—provide its interior" (Baker and Townsend 2007, p. xi). This may be true in much lyric poetry, but in Everett, one argument's interiority might purport to be situated on an exterior "dimensionless plane" (Everett 2015, p. 52), while another idea "might/Find rhythm" (Everett 2015, p. 58) primarily in the poem's form, but that form is further emphasized by the nonlinearity of its rhetorical argument.

What Everett never forgets is that there are not only ideas at play or that they are arranged but the fact of the graphic exterior upon which these ideas play. The difficulty for

readers, as for Everett, is that the graph and ideas are apprehensible only through language; the graph and what it contains are one and the same. The "underlying configuration" for both highly mimetic and experimental poetry, and for Everett, "essentially remains the same basic doubleness of utterance and language, of signified and signifier and the deliberate manipulation of the tensions between the two" (Hühn 1998, p. 218). To buck against this difficulty, Everett's poems foreground the constructedness of language by signaling it as one of many artifices set temporally at play. Surface extends into and pulls back from surface, and it is perceived by surface that is elsewhere and rearranged. The dual emphasis in Everett's first four books of poems on foregrounding constructedness without always unmooring language from meaning helps shield the poetry from what Hühn calls "the risk of self-blockage through the paradoxical, recursive process of self-mirroring" (Hühn 1998, p. 220). These poems' focal breadth allows Everett to mirror perspectival limitations without breaking or being blocked by those same limitations.

**Funding:** This research received no external funding.

**Institutional Review Board Statement:** Not applicable.

**Informed Consent Statement:** Not applicable.

**Data Availability Statement:** Not applicable.

**Conflicts of Interest:** The author declares no conflict of interest.

## Notes

1  I write more about how the hypernarrator in Everett's short fiction resituates the "hyper" in physicalized media in a special issue of *African American Review* about Percival Everett.

2  One might also consider how Everett's work may be in conversation with that of other writers who were or are primarily known for their fiction, as Everett, but who also write or wrote poetry. As is often the case with Everett's work, the intertextual field here is wide, heavily populated, and worth deeper consideration.

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
