# Peer review of "Network Temporality in Percival Everett’s Poetry"

_humanities, doi:10.3390/h12040084_

Round 1

Reviewer 1 Report

To date, there has been little scholarship on Percival Everett's small but growing body of poetry. This essay tackles this relatively small corpus not only through the lens of the lyric poem but in contrast to his own meta-post-postmodern experiments with genre in his fictional output. The writer does a good job displaying the difficult of Everett's poetry, its intertextuality, and its loose relationship with temporality. I think this piece is in publishable shape. My only real suggestion, and it's the kind of thing that could be relegated to a footnote or a parenthetical aside, is to consider how Everett's poetry draws on a tradition of other writers mostly known for their fictional output who also "dabbled" in poetry (like James Joyce, Ishmael Reed, Jack Kerouac, Maxine Hong Kingston, Richard Brautigan, etc.).

Author Response

Thank you very much for your kind, prompt review of my article and for your encouraging response.

Point 1: My only real suggestion, and it's the kind of thing that could be relegated to a footnote or a parenthetical aside, is to consider how Everett's poetry draws on a tradition of other writers mostly known for their fictional output who also "dabbled" in poetry (like James Joyce, Ishmael Reed, Jack Kerouac, Maxine Hong Kingston, Richard Brautigan, etc.).

Response 1: I have included a footnote in the most appropriate place in the essay to incorporate this suggestion.

Reviewer 2 Report

"Network Temporality in Percival Everett's Poetry" will make an important contribution to an understudied aspect of a major contemporary writer. The essay brings a range of new media theoretical work to bear on succinct, insightful, and sometimes brilliant close readings of Everett's often challenging "verse." Further, the questions--what even constitutes poetry--are well posed from the outset and the thesis (hypergraphical network-temporal relations) is crisp and compelling and unfolds into rich and nuanced readings. This eye-opening article demonstrates an easy command of Everett's work, the critical discourse to date, and seems fluent in new media theory (Liu et alia). 

Superb work overall, which I highly recommend.  

The writing is for the most part vivid, thoughtful, and often intricate. Sometimes too intricate? In places it might be too clever for its own good: the opening sentence, for example, piles clause upon clause to layer the very problem of poetry in complex ways: the ideas are nuanced and challenging and I am welcome to such nuanced writing, which is designed to upset our easy understandings (and thus, very much in sync with the content). Stylistically, however, it might help to break the sentence up: it seems overly forbidding as an opening sally. The English here, by the way, is excellent: it is a question of style and rhetoric I am raising, not one of lucidity or clarity.   

Author Response

Thank you so much for prompt review of my article and for your very kind and encouraging response.

Point 1: The writing is for the most part vivid, thoughtful, and often intricate. Sometimes too intricate? In places it might be too clever for its own good: the opening sentence, for example, piles clause upon clause to layer the very problem of poetry in complex ways: the ideas are nuanced and challenging and I am welcome to such nuanced writing, which is designed to upset our easy understandings (and thus, very much in sync with the content). Stylistically, however, it might help to break the sentence up: it seems overly forbidding as an opening sally. The English here, by the way, is excellent: it is a question of style and rhetoric I am raising, not one of lucidity or clarity.   

Response 1: I have taken your suggestion to "break the [first] sentence up," and the opening sally is that much stronger for the change. 

Reviewer 3 Report

This essay's strength lies in its ability to synthesize relevant scholarship with close reading of Everett's poetry, with appropriate problematizing of concepts of author, genre (especially poetry), and meaning. It's well and clearly written. I strongly suggest a revision that makes the introductory portal and the conclusion match more strongly. Including Roof's concept of the Everettian "hypernarrator" is wise, but it's the last thing mentioned as the essay leaps into its body matter. This sets up some confusion for readers who might legitimately expect that the "hypernarrator" will be returned to at the end with some significance. It seems an easy revision that might be accomplished with a simple sentence structure shift so that Roof is subordinated in that last crucial sentence. At the same time, it's possible indeed that being asked to revise this paragraph might lead you to clarify your argument even further in ways that I can't anticipate. At the very least, though, don't set readers up for that expectation, and this strong piece will become even stronger

Author Response

Thank you so much for your prompt review of my essay and your very helpful suggestion.

Point 1: "I strongly suggest a revision that makes the introductory portal and the conclusion match more strongly."

Response 1: I have revised the introduction to incorporate this suggestion. The essay's focus and the expectations that the introduction establishes are much stronger with this change. Thank you.

Reviewer 4 Report

An excellent, elegant and rigorous reading of Everett's work.  The author tackles a difficult writer and does a very fine job of making Everett's work and mission more accessible to someone who may be unfamiliar with the poet.  Really an excellent article.

Author Response

Thank you so much for your prompt and very kind response to my essay.